# Identification and Verification of a Novel MAGI2-AS3/miRNA-374-5p/FOXO1 Network Associated with HBV-Related HCC

**DOI:** 10.3390/cells11213466

**Published:** 2022-11-01

**Authors:** Chao Wang, Kunkai Su, Hanchao Lin, Beini Cen, Shusen Zheng, Xiao Xu

**Affiliations:** 1Key Laboratory of Integrated Oncology and Intelligent Medicine of Zhejiang Province, Department of Hepatobiliary and Pancreatic Surgery, Affiliated Hangzhou First People’s Hospital, Zhejiang University School of Medicine, Hangzhou 310006, China; 2Westlake Laboratory of Life Sciences and Biomedicine, Hangzhou 310024, China; 3NHC Key Laboratory of Combined Multi-Organ Transplantation, Hangzhou 310003, China; 4Institute of Organ Transplantation, Zhejiang University, Hangzhou 310003, China; 5State Key Laboratory for Diagnosis and Treatment of Infectious Diseases, National Clinical Research Center for Infectious Diseases, Collaborative Innovation Center for Diagnosis and Treatment of Infectious Diseases, The First Affiliated Hospital, School of Medicine, Zhejiang University, Hangzhou 310003, China; 6Department of Hepatobiliary and Pancreatic Surgery, The First Affiliated Hospital, Zhejiang University School of Medicine, 79 Qingchun Road, Hangzhou 310003, China

**Keywords:** hepatocellular carcinoma, bioinformatics analysis, ceRNAs, lncRNA MAGI2-AS3/miR-374-5p/FOXO1 network

## Abstract

Background: Hepatocellular carcinoma (HCC) is a very common neoplasm worldwide, and competitive endogenous RNA (ceRNA) plays an important role in the development of HCC. The purpose of this study is to investigate the molecular mechanisms of ceRNAs in HCC. Methods: This study detects potential ceRNAs from HCC through whole genome analysis of lncRNA, miRNA and mRNA expression. We then performed high-throughput sequencing of tissues from five hepatitis B related HCC patients to screen ceRNAs and those screened ceRNAs expressions were verified on tissues from an independent group of six patients. Finally, the function of ceRNAs of interest was illustrated in vitro. Result: Functional and pathway analysis of The Cancer Genome Atlas revealed ceRNA networks. The high-throughput sequencing identified 985 upregulated and 1612 downregulated lncRNAs and 887 upregulated and 1116 downregulated mRNAs in HCC patients. Differentially expressed genes were parallel to cancer-associated processes, comprising 18 upregulated and 35 downregulated significantly enriched pathways including alcoholism and viral carcinogenesis. Among them, a potential ceRNA network was detected and verified in six HCC patients. CeRNAs of the lncRNA MAGI2-AS3/miR-374-5p/FOXO1 pathway were significantly dysregulated in HCC, and validation in vitro showed that FOXO1 is positively regulated by MAGI2-AS3 through the induction of miR-374a/b-5p in HCC cells. In addition, the overexpression of FOXO1 is associated with proliferation, migration, and invasion of HCC cells and increases apoptosis of HCC cells. MiR-374a/b-5p caused an opposite effect by directly suppressing FOXO1 in HCC cells. Conclusion: CeRNA networks were found in HCC and aberrantly expressed ceRNAs of lncRNA MAGI2-AS3/miR-374-5p/FOXO1 plays a crucial role in HCC, assisting in diagnosis and providing a method for treatment.

## 1. Introduction

Hepatocellular carcinoma (HCC) has become one of the most common types of neoplasms in the world, with a high mortality rate [1], which is caused by the hepatitis B virus (HBV) and alcoholic liver disease. HCC has a high incidence in developing countries, especially in east Asia, and it has the second highest incidence and mortality among malignant tumors in adult male patients in China [2]. Studies have shown that HCC progression is caused by abnormal genetic mutations [3]. However, serum markers such as alpha fetoprotein (AFP) [4] have inhibitory effects for playing a role in the early detection of HCC, due to its lack of specificity and sensitivity. The diagnosis of HCC still depends on radiology and biopsy at the early stage of the disease. However, radiology and biopsy are known to have limited sensitivity. Therefore, we should attach great importance to understand the development of HCC and its growth mechanisms and then promote the effectiveness of the diagnosis and treatment regimens.

Recent studies which use microarray technologies and bioinformatics analysis have revealed that noncoding RNAs (ncRNAs), especially long noncoding RNAs (lncRNAs), participate in many biological and pathological processes [5]. This approach can help to identify differentially expressed genes (DEGs) and competitive endogenous RNAs (ceRNAs) which related to the development and progression of HCC. LncRNA is a long non-coding transcript containing 2 to 100,000 nucleotides and has been shown to have various post-transcriptional and translational regulatory functions in living cells [6]. Furthermore, lncRNA has been reported to affect many aspects of cellular homeostasis [7,8]), including proliferation [9], apoptosis, migration, and gene stability [10]. Some functional lncRNAs are tumor specific and may effect the biological function of tumor cells [11] and generate tumor-specific lncRNA biomarkers [12].

In this study, the Cancer Genome Atlas Liver Hepatocellular Carcinoma database (TCGA-LIHC) was used to screen DEGs and construct a ceRNA network. Microarray technologies and bioinformatics analysis were used to verify the related DEGs and microRNA pairs (mRNA/lncRNA) in HCC tissues. Finally, the function of ceRNAs of interest was illustrated in vitro.

## 2. Materials and Methods

### 2.1. CeRNA Network Construction

The R package GDCRNATools (version 1.4.1) was applied to TCGA-LIHC to decipher the correlated miRNA–(mRNA/lncRNA) pairs, and it was also used to create potential ceRNA networks. The most reliable ceRNA triads (*p* < 0.05) were selected for validation.

### 2.2. Patients and Samples

We first chose five hepatitis B-related HCC patients and obtained HCC and para-carcinoma tissue. All HCC tissues were moderately differentiated. Tissue was handled in an RNAlater^®^ Stabilizing solution (Thermo Fisher Scientific, Carlsbad, CA, USA) and stored at ± 80 °C throughout the experiment. Another independent group of six hepatitis B-related HCC patients was used in the validation phase; tumor and para-carcinoma tissues were put in 300 µL RNAiso Plus and stored at −80 °C until used. The clinical information of those samples was collected in Table 1. 

### 2.3. RNA Sample Preparation and Array Hybridization

Trizol reagent was used for the isolation of tissue RNA from the samples (Invitrogen, Waltham, MA, USA), followed by the addition of the Nuclesopine^®^ RNA Purification Kit to the sample, which was designed to purify it. The human LncRNA 4 × 180 K chip customized by Agilent was used to detect the gene expression profile. The total RNA of the sample was detected first, and the Jingxin^®^ Biochip Universal Labeling Kit was used for in vitro amplification and fluorescent labeling. The hybridization image was analyzed and extracted using Agilent feature extraction software (v10.7) and data was normalized and analyzed using Agilent GeneSpring software.

### 2.4. Data Analysis

We used Agilent feature extraction software (v11.0) to analyze the captured array image. The data was analyzed with Agilentgen sampling software. Gene Spring GX v12.1 was used to quantify data. LncRNA and mRNA, with significant expression differences between HCC and para-carcinoma tissues, were identified by fold change and *p*-value. The DEGs were defined as *p*-value < 0.05 and fold change of >1.

### 2.5. Gene Function Annotation

DEGs were assigned to the Gene Ontology (GO) database and Kyoto Encyclopedia of Genes and Genomes (KEGG). The R package clusterProfiler (4.0.2) summarizes and explains the functions and pathway.

### 2.6. RNA Extraction and qRT-PCR Verification

Trizol reagent (Invitrogen, Waltham, MA, USA) was used for the isolation of total RNA from tissues and cells. PCR (Vazyme, Nanjing, China) (for lncRNA and mRNA) and Mir-X miRNA First Chain Synthesis Kit (Takara, Kusatsu, Jan) (for miRNA) were reversed as cDNA using HiScript^®^ II QRT SuperMix. A quantitative real-time polymerase chain reaction (QRT-PCR) was performed in QuantStudio 5 (Applied Biosystems, Waltham, MA, USA) and the dye used was ChamQ Universal SYBR qPCR Master Mix (Vazyme, Nanjing, China). The gene expression was calculated using the 2^−ΔΔΔCt^ method to evaluate the gene expression, using GAPDH (for mRNA and lncRNA) or U6 (for miRNA) as the internal control.

### 2.7. Cell Culture

Internal validation experiments were performed using HepG2 cells from the Chinese Academy of Sciences and the HepG2 was derived from human HCC. Dulbecco’s Modified Eagle’s Medium (DMEM; Gibco, Carlsbad, CA, USA) and 1% penicillin/streptomycin (Beyotime, Nantong, China) were used for culturing cells. Cells were cultivated at 37 °C in an incubator with 5% carbon dioxide.

### 2.8. Cell Transfection

Forkhead Box O1 (FOXO1) overexpression lentiviral vector was constructed and validated by Hanbio Technology Co., Ltd. For lentiviral transfection, Opti-MEM (Invitrogen, Waltham, MA, USA) was used as a medium, and an appropriate amount of viral vector and 5 mg/mL polybrene (Hanbio, Shanghai, China) were added. Puromycin was then used to select stably transfected HepG2 cells.

SiRNA which against FOXO1 was purchased from GenePharma (GenePharma, Shanghai, China), and using si-NC as a control agent, mimics and inhibitors of miR-374a/b-5p were prepared and compared at Sambong Biotech (Sambong Biotech, Beijing, China), and NC mimics and inhibitor-NC were used as controls. All transfections were used under INTERFERin^®^ siRNA and miRNA transfection reagent (Polyplus, New York, NY, USA).

### 2.9. Dual-Luciferase Reporter Assay

The sequences of the full-length MAGI2 antisense RNA 3 (MAGI2-AS3) and FOXO1 binding sites (3′-UTR) and the predicted miR-374a/b-5p binding sites were facilitated by PCR. The wild-type MAGI2-AS3 reporter gene (MAGI2-AS3-wt), wild-type FOXO1 reporter gene (FOXO1-wt), mutated MAGI2-AS3 reporter gene (MAGI2-AS3AS3-mut), and the mutated FOXO1 reporter gene (FOXO1-mut) were constructed by the GeneArt™ Site-Directed Mutagenesis System (Thermo Fisher Scientific, Waltham, MA, USA). The reporters constructed in these two groups transfected miR-NC or miR-374a/b-5p mimics by GP-miRGLO plasmids into HepG2 cells and were then incubated in an incubator for 36 h, followed by the addition of a Dual Lumi™ Dual Luciferase Reporter Assay Kit (Beyotime Biotechnology, Nantong, China) to measure the luciferase activity.

### 2.10. Colony Test

A Cell Counting Kit-8 (CCK-8) measured cell viability. We Seed cells in well plates with 8 × 10^3^ cells/well and then placed the well pates in 37 °C, 5% CO_2_ constant temperature incubator, so that the cells can grow naturally. After adhering to the wall, 10 µL/well CCK-8 reagent (Yeasen, Shanghai, China) was added for 0, 24, 48, 72 h and cells were incubated in an incubator for 1 h. Finally, a SpectraMax iD5 multifunctional microplate reader (Molecular Devices, San Jose, CA, USA) was used for measurement.

### 2.11. Western Blot

After the transfected HepG2 cells were washed with precol phosphate buffer saline (PBS), the cells were lysed with radio-immunoprecipitation assay (RIPA) lysis buffer (Fdbio, Hangzhou, China) and an additional 1% protease inhibitor mixture (Fdbio, Hangzhou, China), and protein extraction was performed. We used the BCA Protein Detection Kit (Fdbio, Hangzhou, China) to detect protein concentrations. Proteins were separated on SDS-PAGE gel (Genscript, Nanjing, China), applied to PVDF film (Millipore, Burlington, VT, USA) and sealed with 5% skim milk powder. FOXO1 (ab179450, Abcam, UK) and GAPDH (ab8245, Abcam, Metropolis, UK) were used to incubate the product. After washing with TBS-Tween 20 buffer, the secondary antibody (Genscript, Nanjing, China) was incubated for 1 h at room temperature. Electro-chemi-luminescence (ECL) solution (Fdbio, Hangzhou, China) was developed using a FluorChem E chemiluminescence gel imaging system (ProteinSimple, Minneapolis, MN, USA). The final results were analyzed using ImageJ software.

### 2.12. Transwell Assay for Migration and Invasion

We used a Transwell chamber (Corning, Somerville, MA, USA) to evaluate the migration and invasion capacities of HepG2 cells. A cell suspension at a concentration of 2.5 × 10^5^ cells/mL was prepared with serum-free medium. We added 200 µL of serum-free cell suspension to the upper chamber and 600 µL of DMEM medium containing 10% fetal bovine serum (FBS) to the lower chamber. After growing in the incubator for 48 h, the chamber was removed. After washing with PBS, fix the cells on the lower side of the insert membrane with methanol for 30 min, and then crystallization with violet staining was performed. Wash the insert in PBS for several seconds to remove excess dye. Cells and the gel in the upper compartment of the insert need to be gently removed by gently wiping the upper side of the membrane with a cotton swab. Dry the insert completely, then mounted, photographed and counted the migrated cells with a microscope (Olympus, Kyoto, Japan). In the cell invasion test, the upper chamber was precoated with Matrigel (BD Biosciences, Franklin Lakes, NJ, USA), and the other steps were same as the migration test.

### 2.13. Flow Cytometry Analysis

Cell apoptosis was detected by an Annexin V-APC/7-AAD apoptosis kit (MultiSciences, Hangzhou, China). Each tube cell sample contained 5 µL Annexin V-APC and 10 µL PI, lightly centrifuged, cultured at room temperature for 5 min in the dark and detected by FACSCalibur flow cytometer (BD Biosciences, Franklin Lakes, NJ, USA).

## 3. Results

### 3.1. Detection and Validation of HCC CeRNA Network

In total, 344 miRNA–(mRNA/lncRNA) pairs of 29 miRNAs were detected from TCGA-LIHC cohort data with a *p*-value < 0.05. Figure 1A showed a panorama of ceRNA prediction, and potential ceRNA triad validation was plotted by microarray in Figure 1B. miRNAs, mRNAs and lncRNAs were dispersed around the miRNAs, and they were colored according to their fold-change direction from our own data in Figure 1B. The lncRNA MAGI2-AS3/miR374-5p/FOXO1 was a part of the HCC ceRNA network shown in Figure 1B.

### 3.2. Identification of Different LncRNAs and mRNAs in HCC Tissues

Differences in lncRNA and mRNA expression were observed in the tissues of HCC patients. A genome-wide analysis found differences in lncRNA and mRNA expression in HCC and para-carcinoma tissue. The 2597 significantly different lncRNAs (log FC ≥ 1.0, *p* < 0.05) consisted of 985 upregulated and 1612 downregulated lncRNAs, as shown in Figure 2A. In addition, 2003 mRNAs were statistically significantly different in expression between the two groups (log FC ≥ 1.0-fold change, *p* < 0.05). In total, the expression of 887 mRNAs was significantly increased and the expression of 1116 mRNAs was significantly decreased. (log FC ≥ 1.0-fold change; *p* < 0.05). A volcano plot illustrated the distribution of data of lncRNAs and mRNAs, as shown in Figure 2B. Heatmaps of the top 700 significant differential expressions of lncRNAs and mRNAs were shown in Appendix A. Appendix A shows the co-expression network of lncRNAs and mRNAs.

### 3.3. Gene Ontology and Pathway Analyses

In the analysis of the cellular component, the nuclear chromosomal part and the chromatin and chromosomal regions were the three most important component processes associated with elevated mRNA, and the three most important component processes associated with mRNA depletion were the extracellular matrix, the side of the membrane, and the external side of the plasma membrane. In molecular function analysis, protein heterodimerization activity, chromatin binding, and DNA catalytic activity were the three main functions in elevated mRNA levels, whereas cofactor binding, tetrapyrrole binding, and heme binding were the three most significant functions of downregulated mRNAs. In the biological process analysis, DNA conformation change, nuclear division, and eDNA packaging were the three most significant processes associated with elevated mRNA; the organic acid biosynthetic process, the carboxylic acid biosynthetic process, and the fatty acid metabolic process were the three most significant processes among downregulated mRNAs (Figure 3 and Figure 4). A pathway analysis identified 18 upregulated and 35 downregulated mRNA target gene pathways, respectively (Figure 5). The alcoholism, viral carcinogenesis, DNA replication, and cell cycle enrichment pathways in upregulated mRNA were important in the occurrence of HCC. In downregulated mRNAs, enrichment pathways including chemical carcinogenesis, fatty acid degradation, and PPAR signaling pathways were associated with the occurrence of HCC.

### 3.4. Validation of lncRNA MAGI2-AS3/miR374-5p/FOXO1 Axis from HCC ceRNA Network

QRT-PCR was applied to validate the ceRNA network of Figure 1B, and the result is shown in Figure 6. We found that miR-374a/b-5p had a high expression in HCC tissue (*p* < 0.05), and lncRNA MAGI2-AS3 had low expression in HCC tissue (*p* < 0.05). mRNA had low expression (*p* < 0.05), including FOXO1, ErbB receptor feedback inhibitor 1 (ERRFI1), and Carbonic anhydrase 2 (CA2). The lncRNA MAGI2-AS3/miR374-5p/FOXO1 network was screened for further experimental validation because it was in line with expectations. StarBase v2.0 suggested that MAGI2-AS3 had binding sites that interacted with miR-374a/b-5p. Our results indicated that the miR-374a/b-5p mimics can reduce the luciferase activity of MAGI2-AS3-wt, but didn’t affect MAGI2-AS3-mut (Figure. 7A). In addition, StarBase v2.0 predicted that FOXO1 might interact with miR-374a/b-5p, and the luciferase reporter showed that miR-374a/b-5p mimics could reduce the luciferase activity of wild-type FOXO1 (Figure 7B). QRT-PCR and western blot results showed that miR-374a/b-5p expression decreased and FOXO1 expression increased when MAGI2-AS3 was overexpressed in HepG2 cells (Figure 7C) and miR-374a/b-5p expression increased and FOXO1 expression decreased when MAGI2-AS3 was suppressed (Figure 7D).

### 3.5. CeRNA Effector FOXO1 Regulates Proliferation, Migration, Invasion, and Apoptosis In Vitro

In order to explore the biological function of FOXO1 on HepG2 cells, we constructed an overexpression lentiviral vector of FOXO1 and designed siRNA to reduce the expression of FOXO1.. QRT-PCR and western blot showed that the expression of FOXO1 mRNA and protein was significantly upregulated in HepG2 cells with the overexpression of FOXO1 (Figure 8A,B). QRT-PCR and western blot also showed that the expression of FOXO1 mRNA and protein was significantly downregulated in HepG2 cells with si-FOXO1 (Figure 8G,H). The results of colony test showed that the OD450nm of HepG2 cells overexpressing FOXO1 decreased significantly, indicating that cell proliferation was inhibited (Figure 8C), while knocking down FOXO1 increased the viability of HepG2 cells (Figure 8I). Flow cytometry analysis showed that the apoptosis rate of HepG2 cells increased from less than 10% to about 14% by the overexpression of FOXO1 (Figure 8D), indicating that the overexpression of FOXO1 promotes HepG2 cell apoptosis. The apoptosis rate of HepG2 cells after knocking down FOXO1 was about 17% and dropped to about 13% when the cells were replaced with serum-free medium and starved for 48 h after 4–6 h of transfection (Figure 8J), suggesting that cell apoptosis was inhibited. We found that after overexpression of FOXO1, the number of migrating cells in HepG2 cells decreased from about 30 to fewer than 10 through the Transwell test (Figure 8E), while the number of invading cells decreased from nearly 80 to about 30 (Figure 8F), indicating that the migration and invasion ability of HepG2 cells was inhibited. Meanwhile, knocking down FOXO1 showed the opposite effect; si-FOXO1 increased migration and invasion ability compared with the NC group (Figure 8K,L). We then repeated those experiments in the LM3 cell line and found a similar result (Figure 9). Taken together, these results indicate that FOXO1 exerts a tumor-suppressing effect in HCC cells.

### 3.6. MiR-374A-5P and MiR-374B-5P Regulate the Effect of FOXO1 In Vitro

The mRNA and protein of FOXO1 overexpressing HepG2 cells combined with miR-374a-5p and miR-374b-5p mimics significantly reduced the expression of FOXO1 compared with FOXO1 overexpression with NC mimic treated cells (Figure 10A,B). The FOXO1 expression of HepG2 cells transfected with si-FOXO1 was significantly increased due to co-transfection of inhibitor-374a-5p or inhibitor-374b-5p (Figure 10A,B). The colony test showed that the viability of proliferation of FOXO1 overexpressing HepG2 cells combined with transfecting miR-374a/b-5p mimics was significantly increased compared with the NC group. The inhibitory effect of FOXO1 on the proliferation of HepG2 cells was eliminated (Figure 10C). Similarly, transfection of inhibitor-374a-5p and inhibitor-374b-5p offset the proliferation-promoting effect of knocking down FOXO1 on HepG2 cells (Figure 10G).

Flow cytometry analysis showed that the apoptosis rate of the stable HepG2 cell line overexpressing FOXO1 decreased from about 22% to about 18%, and due to the transfecting miR-374a/b-5p, mimics decreased to about 14% (Figure 10D). In addition, after 48 h of culture in serum-free medium, the apoptotic rate of HepG2 cells transfected with si-FOXO1 was significantly increased due to co-transfection of inhibitor-374a/b-5p (Figure 10H), suggesting that the FOXO1 effect on HepG2 cell apoptosis was offset by miR-374a/b-5p.

The Transwell test showed that transfection of miR-374a/b-5p mimics offset the suppression of FOXO1 overexpression on the migration and invasion of HepG2 cells (Figure 10E,F). Transfection of inhibitor-374a-5p and inhibitor-374b-5p offset the promoting effect of FOXO1 knockdown on HepG2 cell migration and invasion (Figure 10I,J). We then repeated those experiments on the LM3 cell line and found similar results (Figure 11).

## 4. Discussion

HCC is a heterogeneous disease, with different histopathological features and clinical behavior between different subtypes. LncRNA was receiving increasing attention to cancer due to its abnormal expression among numerous molecules which were shown to play a role in HCC.

In the present study, we provided novel targets among the many as yet uncharacterized lncRNAs and performed an analysis to identify potential key genes correlated with HCC. Indeed, the 2597 significantly differentially expressed (f ≥ 1.0-fold change) lncRNAs and 2003 mRNAs contained abundant information worthy of further study.

GO analysis provided a shortcut to describe the characteristics of gene products along with functional annotation data such as cell part, cellular process, and catalytic activity. Alcoholism and metabolic pathways were significantly enriched in the KEGG terms. We found that the significantly enriched terms derived from KEGG analysis agreed with the GO results. Among the GO enrichments and KEGG pathways, cell- and metabolic-related pathways appear to play vital roles in HCC carcinogenesis and progression.

The R package GDCRNATools (version 1.4.1) used in this study was applied to the TCGA-LIHC cohort to decipher relevant lncRNA-miRNA-mRNA pairs to construct a potential ceRNA network. A bioinformatics analysis identified potential key genes related to the occurrence and development of HCC, and then PCR verification found that genes such as FOXO1, ERRFI1 and CA2 had low expression in HCC tissue, indicating that these genes might play key roles in the occurrence and development of HCC. ERRFI1 is currently recognized as an immediate early response gene. The product of the ERRFI1 gene was readily induced by a variety of extracellular and intracellular stimuli [13]. ERRFI1 was located on chromosome 1p36 in humans and a locus that has long been thought to contain many putative tumor suppressor genes, and was frequently altered in many cancers including HCC [14]. CA2 was an isoform of the CA family, and the protein encoded by the gene was one of several co-acting enzymes of carbonic anhydrase and played an important role in the reversible hydration–dehydration of carbon dioxide. This gene had been found to encode two transcript variants of different isoforms. The role of CA2 in cancer depended on the type of tumor. For example, if CA2 is overexpressed, it could be used as a powerful biomarker for the diagnosis of gastrointestinal matrix tumors (GIST) [15]. However, CA2 inhibited the growth of cancer cells in colorectal cancer [16]. In bladder cancer, CA2 promoted tumor growth [17]. However, current research shows that CA2 expression is downregulated in HCC tissue and was associated with HCC growth and metastasis [18].

The lncRNA-miRNA-mRNA regulatory network was determined by bioinformatics and microarray and verified the results by qRT-PCR of HCC tissues. MAGI2-AS3/miRNA-374-5p/FOXO1 was in line with our expectation among those network, so we continued to study the lncRNA MAGI2-AS3/miRNA-374-5p/mRNA FOXO1 pathway. Even though part of the ceRNA of lncRNA MAGI2-AS3-miRNA-374-5p has been verified by related studies in HCC [19], we researched the regulatory axis from other perspectives. The signaling axis of miR-374a/FOXO1 was first verified in ovarian cancer, and miR-374a was found to modulate FOXO1 to control the proliferation in ovarian cancer cells [20]. FOXO1 is an important transcription factor in insulin signaling and an important repressor in PI3K/Akt signaling, which regulates various biological processes, such as cell cycle, cell differentiation, tumorigenesis, and oxidative stress responses [21]. In this study, we investigated the role of this pathway in HCC and found that it could regulate biological behaviors such as the proliferation, invasion and apoptosis of HCC cells.

It was first time we confirmed that MAGI2-AS3 could modulate FOXO1 expression and miR-374a/b-5p play a tumor-promoting role by modulating FOXO1 in HCC. FOXO1 is a tumor-suppressing factor that inhibits carcinogenesis [22]. Disruption of FOXO1 level/activity promoted carcinogenesis. MiR-374a/b-5p could bind to FOXO1 and negatively regulated its expression, which resulted in promoting carcinogenesis in HCC.

In our study, we found that the ceRNAs expressed in HCC tissue were significantly different compared to para-carcinoma tissue. However, the role of these genes in HCC is not clear now. With regard to a subsequent pathway, studies should be focused on the differentially expressed ceRNAs. Our study revealed a novel molecular mechanism of HCC cells caused by the lncMAGI2-AS3/miR-374-5p/FOXO1 pathway in vitro. Further investigations are required to deeply understand this complex mechanism.

## 5. Conclusions

Our bioinformatics analysis identifies a range of lncRNAs and mRNAs, which included the differential expression in HCC and para-carcinoma tissues. These DEGs constructed a ceRNA network and provided novel targets for studying HCC pathways and mechanisms, as well as new targets for developing diagnostic and therapeutic methods. Furthermore, we demonstrated that miR-374a-5p and miR-374b-5p negatively regulate FOXO1 expression to modulate the biological behavior of HCC cells. These new findings showed that miR-374a/b-5p could be effective targets for the treatment of HCC.

## Figures and Tables

**Figure 1 cells-11-03466-f001:**
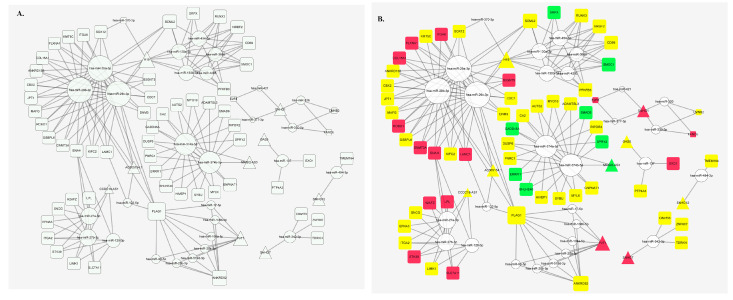
CeRNAs regulatory network. (**A**) ceRNA regulatory network predicted by TCGA. (**B**) potential ceRNA network validation is plotted by microarray. Blank circles represent predicted miRNAs. The green square represents down-regulated mRNA that we detected with a *p* value < 0.05, the yellow square represents mRNA that we detected with *p* value > 0.05, and the red square represents up-regulated mRNA that we detected with *p* value < 0.05. The green triangle represents down-regulated lncRNA that we detected with *p* value < 0.05, the yellow triangle represents lncRNA that we detected with *p* value > 0.05, and the red triangle represents up-regulated lncRNA that we detected with *p* value < 0.05. ceRNA, competitive endogenous RNA. TCGA, the Cancer Genome Atlas.

**Figure 2 cells-11-03466-f002:**
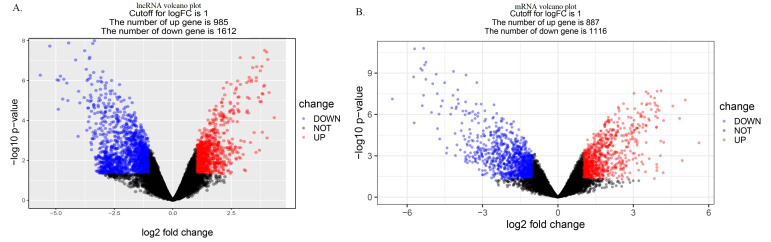
Volcano plots illustrating data distributions in (**A**) lncRNA and (**B**) mRNA profiles. Red denotes high relative expression and blue denotes low relative expression.

**Figure 3 cells-11-03466-f003:**
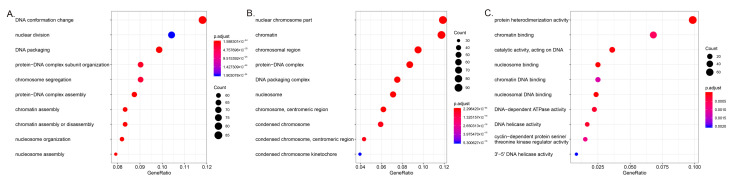
Gene Ontology enrichment analysis of upregulated mRNA between hepatocellular carcinoma and para-carcinoma tissues: (**A**) biological process, (**B**) cell component, (**C**) molecular function.

**Figure 4 cells-11-03466-f004:**
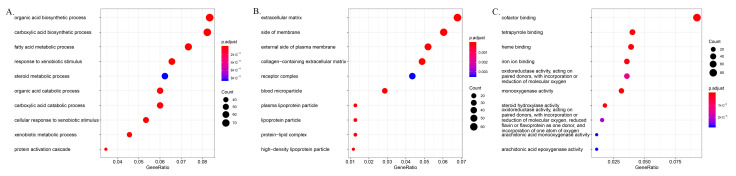
Gene Ontology enrichment analysis of downregulated mRNA between hepatocellular carcinoma and para-carcinoma tissues: (**A**) biological process, (**B**) cell component, (**C**) molecular function.

**Figure 5 cells-11-03466-f005:**
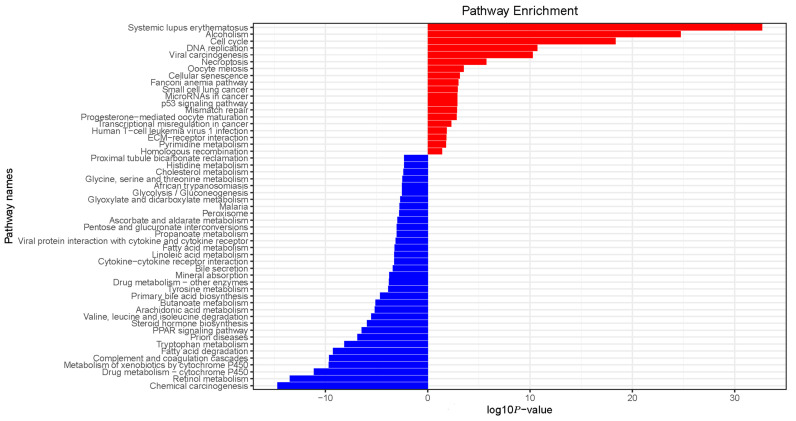
Top pathways associated with upregulated and downregulated pathways of mRNAs.

**Figure 6 cells-11-03466-f006:**
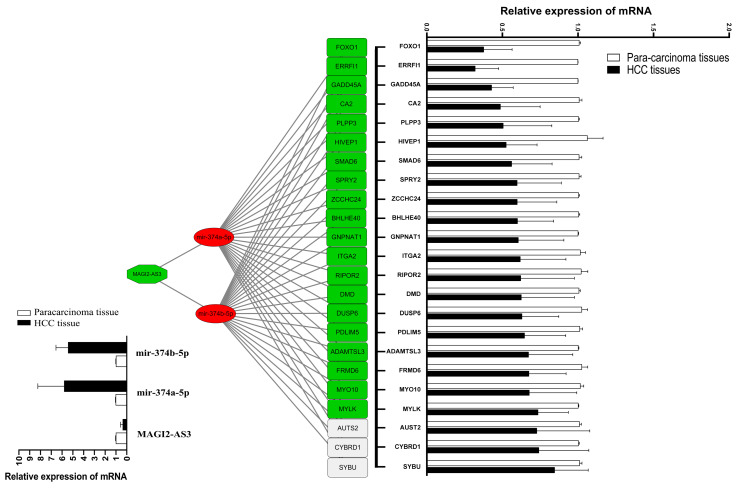
CeRNA network by HCC tissue sequencing predicted by qRT-PCR. The red circle represents upregulated miRNAs detected with *p*-value < 0.05 and verified, green square represents downregulated mRNAs detected with *p*-value < 0.05 and verified, green hexagon represents downregulated lncRNAs detected with *p*-value < 0.05 and verified. Blank square represents downregulated mRNAs detected with *p*-value < 0.05 but verified with *p*-value > 0.05. Relative expression levels of lncRNAs, miRNAs, and mRNA verified by qRT-PCR are shown on the left and right sides. HCC, hepatocellular carcinoma.

**Figure 7 cells-11-03466-f007:**
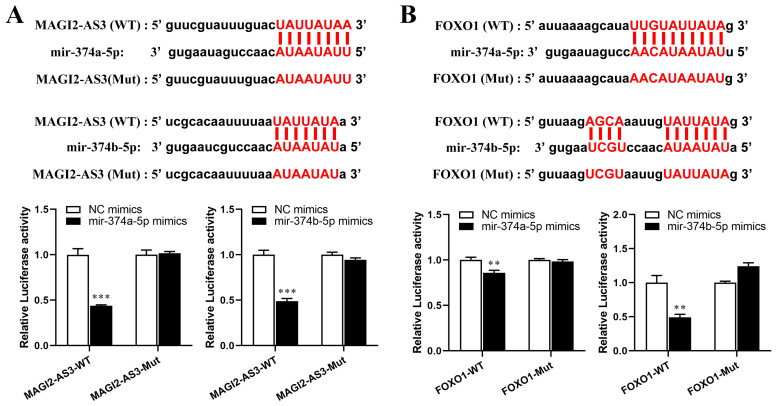
(**A**) Sequences of MAGI2-AS3/miR-374a/b-5p binding site predicted by StarBase 2.0. MiR-374a/b-5p mimics only reduced MAGI2-AS3-wt luciferase activity. (**B**) Sequences of miR-374a/b-5p /FOXO1 binding site predicted by StarBase 2.0. MiR-374a-5p/miR-374b-5p mimics only reduced FOXO1-wt luciferase activity. (**C**) QRT-PCR shows that MAGI2-AS3 and FOXO1 expression were increased and miR-374a/b-5p was decreased, and western blot results showed that FOXO1 protein level was increased due to overexpression of MAGI2-AS3. (**D**) QRT-PCR results showed that mRNA levels of MAGI2-AS3 and FOXO1 expression were decreased and miR-374a/b-5p was increased, and western blot results showed that the FOXO1 protein level was decreased due to the inhibition of MAGI2-AS3. * *p* < 0.05, ** *p* < 0.01, *** *p* < 0.001. MAGI2-AS3, MAGI2 antisense RNA 3. WT, wild type. Mut, mutant. FOXO1, Forkhead Box O1.QRT-PCR, quantitative real-time polymerase chain reaction.

**Figure 8 cells-11-03466-f008:**
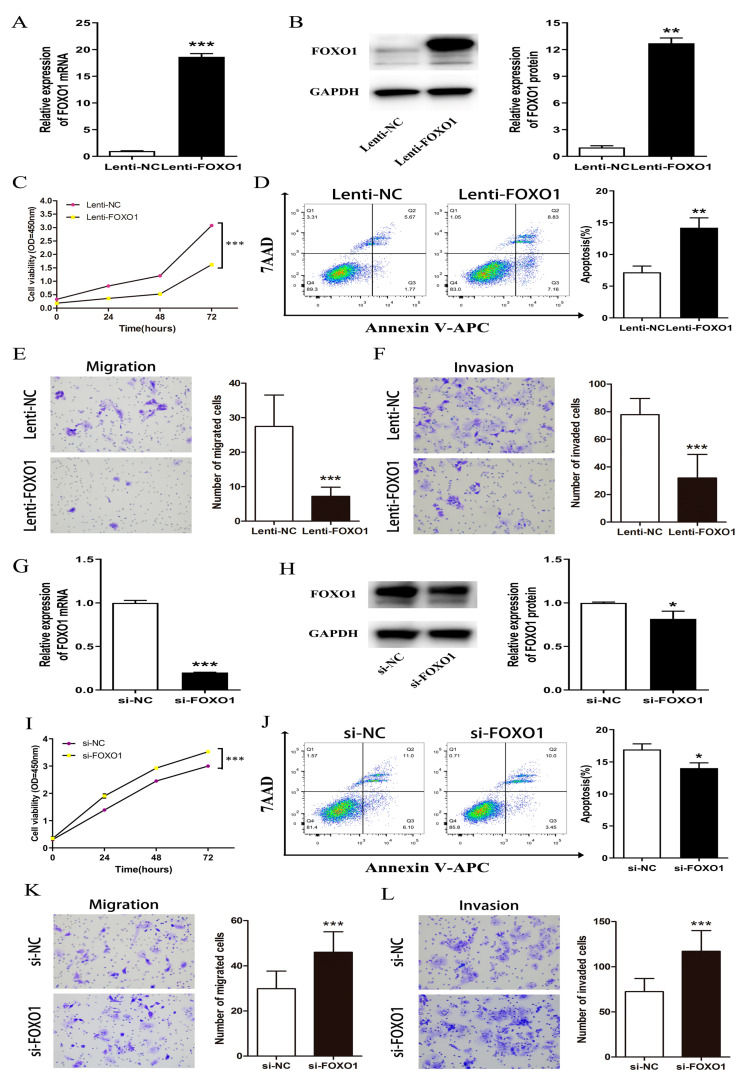
(**A**,**B**) QRT-PCR and western blot showed that expression of FOXO1 mRNA and protein were significantly upregulated in HepG2 cells transfected with overexpressed vector. (**C**) Cell proliferation was decreased significantly in HepG2 cells by overexpressing FOXO1; (**D**) A flow cytometry analysis indicated increased apoptosis induced by FOXO1 overexpression in HepG2 cells. (**E**,**F**) A Transwell test revealed the overexpression of FOXO1 suppressed HepG2 cell migration and invasion ability induced by MAGI2-AS3 overexpression in HepG2 cells. (**G**,**H**) QRT-PCR and western blot showed that the expression of FOXO1 mRNA and proteins were significantly downregulated in HepG2 cells with si-FOXO1. (**I**) HepG2 cells with si-FOXO1 had increased viability of proliferation. (**J**) Flow cytometry analysis indicated decreased apoptosis of HepG2 cells after knocking down FOXO1. (**K**,**L**) A transwell test revealed that migration and invasion ability increased in HepG2 cells by knocking down FOXO1. * *p* < 0.05, ** *p* < 0.01, *** *p* < 0.001. FOXO1, Forkhead Box O1. QRT-PCR: quantitative real-time polymerase chain reaction.

**Figure 9 cells-11-03466-f009:**
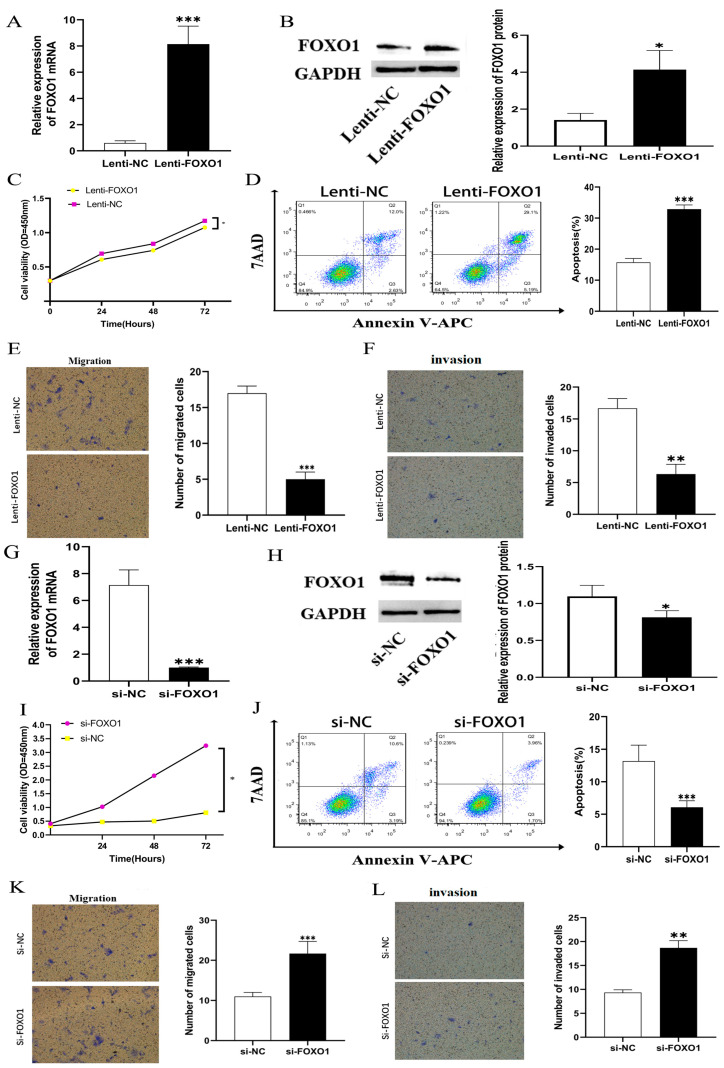
(**A**,**B**) QRT-PCR and western blot showed that expression of FOXO1 mRNA and protein were significantly upregulated in LM3 cells transfected with overexpressed vector. (**C**) Cell proliferation was decreased significantly in LM3 cells by overexpressing FOXO1; (**D**) A flow cytometry analysis indicated increased apoptosis induced by FOXO1 overexpression in LM3 cells. (**E**,**F**) A transwell test revealed overexpression of FOXO1 suppressed LM3 cell migration and invasion ability induced by MAGI2-AS3 overexpression in LM3 cells. (**G**,**H**) QRT-PCR and western blot showed that expression of FOXO1 mRNA and protein were significantly downregulated in LM3 cells with si-FOXO1. (**I**) LM3 cells with si-FOXO1 had increased viability of proliferation. (**J**) Flow cytometry analysis indicated decreased apoptosis of LM3 cells after knocking down FOXO1. (**K**,**L**) A Transwell test revealed that migration and invasion ability increased in LM3 cells by knocking down FOXO1. * *p* < 0.05, ** *p* < 0.01, *** *p* < 0.001. FOXO1, Forkhead Box O1. QRT-PCR: quantitative real-time polymerase chain reaction.

**Figure 10 cells-11-03466-f010:**
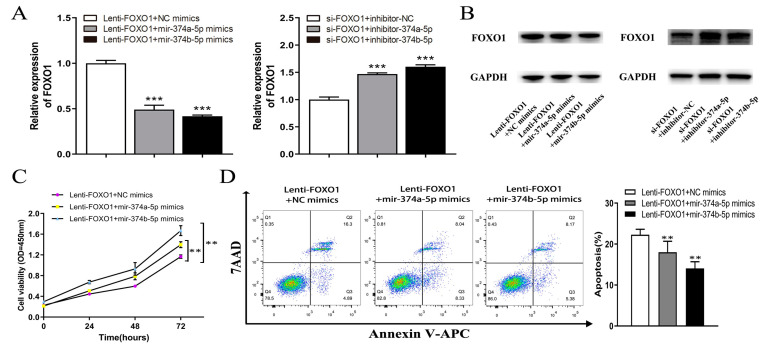
(**A**) the mRNA of FOXO1 overexpression in HepG2 cells combined with miR-374a/b-5p mimics significantly reduced FOXO1 expression compared with FOXO1 overexpression with NC mimic treated cells, and the mRNA of FOXO1 expression in HepG2 cells transfected with si-FOXO1 was significantly increased due to co-transfection of inhibitor-374a-5p or inhibitor-374b-5p. (**B**) The protein of FOXO1 overexpression in HepG2 cells combined with miR-374a/b-5p mimics significantly reduced FOXO1 expression compared with FOXO1 overexpression with NC mimic treated cells, and the mRNA of FOXO1 expression in HepG2 cells transfected with si-FOXO1 was significantly increased due to co-transfection of inhibitor-374a-5p or inhibitor-374b-5p. (**C**) FOXO1 overexpression combined with miR-374a/b-5p mimics significantly increased the viability of proliferation of HepG2 cells compared with FOXO1 with NC mimic treated HepG2 cells. (**D**) The apoptosis rate of stably transfected HepG2 cell lines overexpressing FOXO1 was reduced by transfection of miR-374a/b-5p mimics. (**E**,**F**) A transwell test showed that transfection of miR-374a/b-5p mimics the offset effect of suppressing FOXO1 overexpression on migration and invasion of HepG2 cells. (**G**) Transfection of inhibitor-374a-5p and inhibitor-374b-5p offset the proliferation-promoting effect of knocking down FOXO1 on HepG2 cells. (**H**) The apoptosis rate of HepG2 cells transfected with si-FOXO1 was significantly increased due to the co-transfection of inhibitor-374a-5p or inhibitor-374b-5p. (**I**,**J**) Transfection of inhibitor-374a-5p and inhibitor-374b-5p offset promoting effect of FOXO1 knockdown on HepG2 cell migration and invasion. * *p* < 0.05, ** *p* < 0.01, *** *p* < 0.001. FOXO1, Forkhead Box O1. QRT-PCR: quantitative real-time polymerase chain reaction.

**Figure 11 cells-11-03466-f011:**
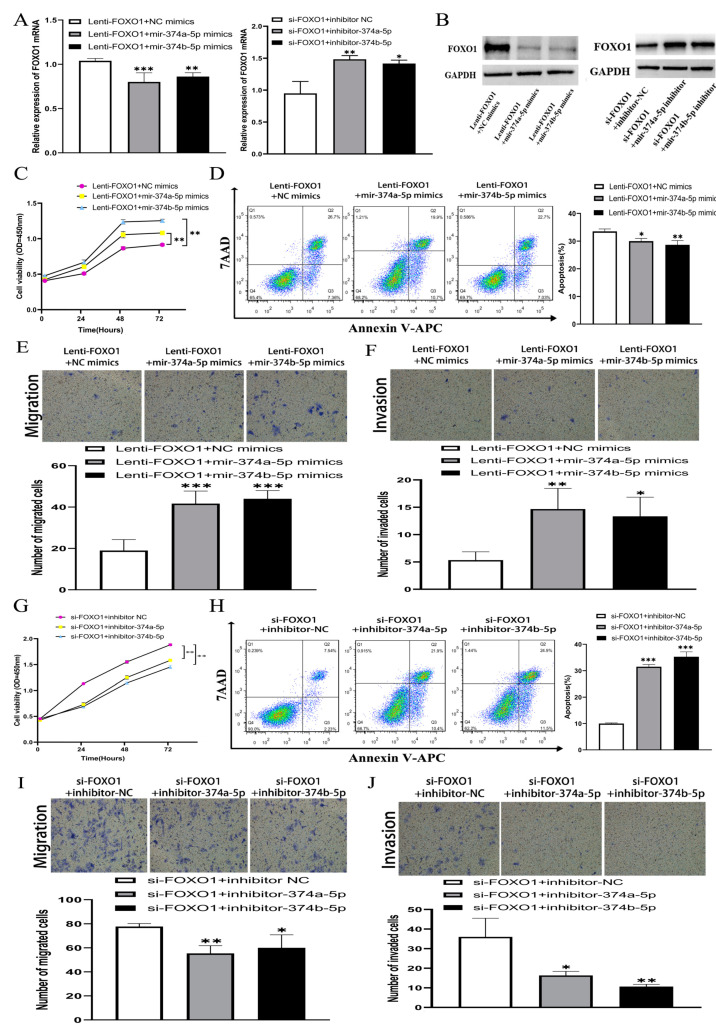
(**A**) the mRNA of FOXO1 overexpression in LM3 cells combined with miR-374a/b-5p mimics significantly reduced FOXO1 expression compared with FOXO1 overexpression with NC mimic treated cells, and the mRNA of FOXO1 expression in LM3 cells transfected with si-FOXO1 was significantly increased due to co-transfection of inhibitor-374a-5p or inhibitor-374b-5p. (**B**) The protein of FOXO1 overexpression in LM3 cells combined with miR-374a/b-5p mimics significantly reduced FOXO1 expression compared with FOXO1 overexpression with NC mimic treated cells, and the mRNA of FOXO1 expression in LM3 cells transfected with si-FOXO1 was significantly increased due to the co-transfection of inhibitor-374a-5p or inhibitor-374b-5p. (**C**) FOXO1 overexpression combined with miR-374a/b-5p mimics significantly increased the viability of proliferation of LM3 cells compared with FOXO1 with NC mimic treated LM3 cells. (**D**) The apoptosis rate of stably transfected LM3 cell lines overexpressing FOXO1 was reduced by transfection of miR-374a/b-5p mimics. (**E**,**F**) A transwell test showed that transfection of miR-374a/b-5p mimics offset the effect of suppressing FOXO1 overexpression on the migration and invasion of LM3 cells. (**G**) The transfection of inhibitor-374a-5p and inhibitor-374b-5p offset proliferation-promoting effect of knocking down FOXO1 on LM3 cells. (**H**) The apoptosis rate of LM3 cells transfected with si-FOXO1 was significantly increased due to the co-transfection of inhibitor-374a-5p or inhibitor-374b-5p. (**I**,**J**) Transfection of inhibitor-374a-5p and inhibitor-374b-5p offset promoting effect of FOXO1 knockdown on LM3 cell migration and invasion. * *p* < 0.05, ** *p* < 0.01, *** *p* < 0.001. FOXO1, Forkhead Box O1. QRT-PCR: quantitative real-time polymerase chain reaction.

**Table 1 cells-11-03466-t001:** Clinical characteristics of samples.

Sample Name	Sex	Age	Tumor Size	Histological Grading	AFP (ug/L)	HBV-DNA (IU/mL)	HbsAg
Sample 1	Male	48	3.5 cm × 5 cm	Moderately differentiated	556	1.75 × 10 e^4^	Positive
Sample 2	Male	53	3 cm × 3.5 cm	Moderately differentiated	875	<20	Positive
Sample 3	Male	49	2 cm × 3 cm	Moderately differentiated	310	<20	Positive
Sample 4	Male	54	2.5 cm × 3 cm	Moderately differentiated	228	1.49 × 10 e^2^	Positive
Sample 5	Male	37	3.5 cm × 6 cm	Moderately differentiated	448	<20	Positive
Sample 6	Male	56	3 cm × 4 cm	Moderately differentiated	621	<20	Positive
Sample 7	Male	69	2 cm × 5 cm	Moderately differentiated	355	<20	Positive
Sample 8	Male	64	4 cm × 5 cm	Moderately differentiated	746	<20	Positive
Sample 9	Male	57	3 cm × 3.5 cm	Moderately differentiated	583	<20	Positive
Sample 10	Male	53	3 cm × 5 cm	Moderately differentiated	173	1.38 × 10 e^2^	Positive
Sample 11	Male	49	2 cm × 2.5 cm	Moderately differentiated	279	<20	Positive

AFP, alpha-fetoprotein. HBV, hepatitis B virus.

## Data Availability

The data generated in the present study may be requested from the corresponding author.

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
