# Peer review of "Identification and Verification of a Novel MAGI2-AS3/miRNA-374-5p/FOXO1 Network Associated with HBV-Related HCC"

_cells, 2022, doi:10.3390/cells11213466_

Round 1

Reviewer 1 Report

This study used bioinformation and high-throughput sequencing of HBV-related HCC patients to MAGI2-AS3/miRNA-374-5p/FOXO1 network and confirm it by HepG2 cells. However, some questions need to solve. 

1. siRNA-induced gene knockdown has inevitable off-target effects. The author should use two or more siRNAs to verify the off-target effects. In addition, the author should present data in at least 2 HCC cell lines to extend their study.

2. This study discussed HBV-related HCC, why did the authors use HepG2 cells?

3. The sequences of siRNA or microRNA mimic/inhibitor need to expose.

4. In abstract, "high-throughput sequencing of 10 samples from 5 HBV-related HCC patients". However, there are 11 HBV-related HCC patients in table 1. Which one is right?

5.  This manuscript has text similarity with previously published sources. The authors should need to modify.  

Author Response

This study used bioinformation and high-throughput sequencing of HBV-related HCC patients to MAGI2-AS3/miRNA-374-5p/FOXO1 network and confirm it by HepG2 cells. However, some questions need to solve. 

  1. siRNA-induced gene knockdown has inevitable off-target effects. The author should use two or more siRNAs to verify the off-target effects. In addition, the author should present data in at least 2 HCC cell lines to extend their study.

Response: Thank you for your good comments. It is indeed that siRNA-induced gene knockdown will have off-target effects, so we used three siRNA to knockdown FOXO1(showed in response 3) and validate the effect of siRNA by RT-qPCR on the level of transcription and western blot on the level of translation. The result of RT-qPCR and western blot showed the expression of FOXO1 was downregulated by siRNA (figure 8).

We will follow your guidance about using another cell line to further verify the conclusion, and compare it with the current research, some research about the MAGI2-AS3/miRNA-374-5p/FOXO1 is undergoing. We will ask editor for prolong the deadline to finish the experiment in vitro.

  1. This study discussed HBV-related HCC, why did the authors use HepG2 cells?

Response: The cell line has a high degree of differentiation, stable metabolic enzymes, and will not change due to the increase in the number of passages, so it is an reasonable cell line for the study of HCC. Furthermore, we are using HepG2.2.15 in the follow-up research. HepG2.2.15 are derived from HepG2 cells and stably transfected with HBV gene, which should be more ideal to mimic the pathology of interest in our study.

  1. The sequences of siRNA or microRNA mimic/inhibitor need to expose.

Response: the sequences exposed as follows,

sense(5'-3')

antisense(5'-3')

miR-374a-5p mimics

UUAUAAUACAACCUGAUAAGUG

CUUAUCAGGUUGUAUUAUAAUU

miR-374b-5p mimics

AUAUAAUACAACCUGCUAAGUG

CUUAGCAGGUUGUAUUAUAUUU

miR-374a-5p inhibitor

CACUUAUCAGGUUGUAUUAUAA

miR-374b-5p inhibitor

CACUUAGCAGGUUGUAUUAUAU

FOXO1-siRNA

CCCUCGAACUAGCUCAAAUTT

AUUUGAGCUAGUUCGAGGGTT

FOXO1-siRNA

CCCAGUCUGUCUGAGAUAATT

UUAUCUCAGACAGACUGGGTT

FOXO1-siRNA

CUGCAUCCAUGGACAACAATT

UUGUUGUCCAUGGAUGCAGTT

  1. In abstract, "high-throughput sequencing of 10 samples from 5 HBV-related HCC patients". However, there are 11 HBV-related HCC patients in table 1. Which one is right?

Response: Thank you for your question. we used 10 samples from 5 HBV-related HCC patients for high-throughput sequencing to construct ceRNA network and12 samples from 6 HBV-related HCC patients including HCC and para-carcinoma samples for ceRNA network validation by PT-qPCR. We added some content in abstract to make it more clearly.

  1. This manuscript has text similarity with previously published sources. The authors should need to modify.  

Response: the manuscript was checked by iThenticate/CrossCheck and the showed 16% similarity index, we uploaded the report as an attachment. Most of the similarity comes from Materials and Methods and we modified the content of this section.

Reviewer 2 Report

In the present article, the authors identified a potential new network associated with HBV-related HCC. However, a few points should be taken into consideration:

1. The english writting should be improved and spelling mistakes redone, specifically in the materials and methods section. Additionally, the chosen scientific wording should be rethinked; for example: "Transfected cells were digested...". Not only are the cells not "digested", but also "digested" is not the scientific word to be used.

 2. Throughout the article the authors used, in some cases, the abbreviations to designate a molecule without explaining its meaning prior in the text; while in other cases they do not mention these abbreviations at all. Additionally, the authors should standardize the used abbreviations; for example, in the materials and methods section, the word "minute" is written as "minute" and as "min".

3. The authors wrote "µL" as "uL". This is not acceptable.

4. In table 1, the legend for "AFP" more perceptive to the reader.

5. In section 3.4 the title reads as "In vitro validation of lncRNA MAGI2-AS3/miR374-5p/FOXO1 ceRNA axis". However, it is not very perceptive for the reader from the previous sections how the authors came to this selection of molecules.

6. In the results section, it is mentioned a colony test. However, it is not clear for the reader to which method this refers to.

7. In the discussion section, the authors discussed more about the genes already identified for HCC than the ones obtained.

Author Response

In the present article, the authors identified a potential new network associated with HBV-related HCC. However, a few points should be taken into consideration:

  1. The english writting should be improved and spelling mistakes redone, specifically in the materials and methods section. Additionally, the chosen scientific wording should be rethinked; for example: "Transfected cells were digested...". Not only are the cells not "digested", but also "digested" is not the scientific word to be used.

Response: We apologize for the poor language of our manuscript. We worked on the manuscript for a long time and the repeated addition and removal of sentences and sections obviously led to poor readability. We have now worked on both language and readability and have polished the manuscript by an English language editing company. We really hope that the flow and language level have been substantially improved.

  1. Throughout the article the authors used, in some cases, the abbreviations to designate a molecule without explaining its meaning prior in the text; while in other cases they do not mention these abbreviations at all. Additionally, the authors should standardize the used abbreviations; for example, in the materials and methods section, the word "minute" is written as "minute" and as "min".

Response: We are very sorry for our incorrect writing of abbreviations, the statements of “minute” were corrected as “min”

  1. The authors wrote "µL" as "uL". This is not acceptable.

Response: We are very sorry for our incorrect writing, the statements of “uL” were corrected as “µL”

  1. In table 1, the legend for "AFP" more perceptive to the reader.

Response: Alpha-fetoprotein is the legend for “AFP”. I am a little confused about the comment, can you make it more clearly?

  1. In section 3.4 the title reads as "In vitro validation of lncRNA MAGI2-AS3/miR374-5p/FOXO1 ceRNA axis". However, it is not very perceptive for the reader from the previous sections how the authors came to this selection of molecules.

Response: Bioinformatics and microarray were used to construct the ceRNA network of HCC and the result of the network was shown in section 3.1 firstly. the lncRNA MAGI2-AS3/miR374-5p/FOXO1 ceRNA axis was showed in section 3.1 as a part of the ceRNA network. PCR was used to preliminary verification about the ceRNA network and then we found the expression level of lncRNA MAGI2-AS3/miR374-5p/FOXO1 axis was accord with expectation of the study. Therefore, we choose the axis to explore the biological function. We have added some content in materials and methods and results section and change the title of 3.4 section as "Validation of lncRNA MAGI2-AS3/miR374-5p/FOXO1 axis from HCC ceRNA network" to make it more clearly.

  1. In the results section, it is mentioned a colony test. However, it is not clear for the reader to which method this refers to.

Response: Cell Counting Kit-8 (CCK-8) assay is one of the methods to test the viability of cells. we change “2.10. Cell Counting Kit-8 (CCK-8) assay” to “2.10. colony test” and revise the content of 2.10 section to make it more clearly.

  1. In the discussion section, the authors discussed more about the genes already identified for HCC than the ones obtained.

Response: Thank you for your comment. The genes discussed in the discussion section were all detected in microarray and preliminary validated by RT-qPCR in HCC tissues. We agreed with you and deleted some irrelevant genes content in discussion section.

Reviewer 3 Report

97 / 5.000  

Risultati della traduzione

The work looks interesting. It is useful to compare with other groups with different aetiology of HCC

Author Response

response: thank you for your precious suggestion, we will follow your guidance and compare it with the current research.

Round 2

Reviewer 1 Report

I agree with the publishment of this manuscript.

Author Response

Thank you very much for your precious comments!